# Unsupervised Learning of Efficient Exploration: Pre-training Adaptive Policies via Self-Imposed Goals

**Octavio Pappalardo** *
Independent Researcher
octaviopappalardo@gmail.com

## Abstract

Unsupervised pre-training can equip reinforcement learning agents with prior knowledge and accelerate learning in downstream tasks. A promising direction, grounded in human development, investigates agents that learn by setting and pursuing their own goals. The core challenge lies in how to effectively generate, select, and learn from such goals. Our focus is on broad distributions of downstream tasks where solving every task zero-shot is infeasible. Such settings naturally arise when the target tasks lie outside of the pre-training distribution or when their identities are unknown to the agent. In this work, we (i) optimize for efficient multi-episode exploration and adaptation within a meta-learning framework, and (ii) guide the training curriculum with evolving estimates of the agent's post-adaptation performance. We present ULEE, an unsupervised meta-learning method that combines an in-context learner with an adversarial goal-generation strategy that maintains training at the frontier of the agent's capabilities. On XLand-MiniGrid benchmarks, ULEE pre-training yields improved exploration and adaptation abilities that generalize to novel objectives, environment dynamics, and map structures. The resulting policy attains improved zero-shot and few-shot performance, and provides a strong initialization for longer fine-tuning processes. It outperforms learning from scratch, DIAYN pre-training, and alternative curricula. Code is available at: https://github.com/Octavio-Pappalardo/ulee-jax

## 1 Introduction

Large-scale pre-training has underpinned many real-world successes of deep learning in computer vision (Krizhevsky et al., 2012; Chen et al., 2020; Grill et al., 2020; He et al., 2022) and language modeling (Devlin, 2018; Brown, 2020). These advancements have motivated analogous efforts in deep reinforcement learning (RL) (Liu & Abbeel, 2021; Schwarzer et al., 2021; Seo et al., 2022), where the prevailing paradigm remains to train agents from scratch for each new task. The ultimate objective is to develop foundation policies: agents equipped with transferable knowledge that address RL's fundamental challenges of sample inefficiency and lack of generalization. We investigate unsupervised RL, where agents interact freely with their environments without access to extrinsic rewards, as a means to acquire such policies. This setting raises key questions regarding what data an agent should seek and what it should learn from it. These questions are tightly coupled because the collected data is a direct consequence of the agent's incorporated behaviors. Ultimately, we seek a system that can continuously gather useful data at scale in an unsupervised, open-ended fashion and leverage it to acquire transferable capabilities.

A promising direction is to design agents that autonomously generate their own goals and learn by attempting to solve them (Oudeyer et al., 2007; Nair et al., 2018; Forestier et al., 2022). Our focus is on how such goals should be generated, selected, and exploited for pre-training. This connects to automatic curriculum learning, yet existing methods have often assumed either a fixed set of goals or a goal space that is closely aligned with the evaluation tasks. Progress toward foundation policies

---

*Work conducted as an independent researcher. Currently at University College London (UCL).

requires assuming little or no information about downstream objectives and instead pre-training for a broad range of human-relevant tasks. Moreover, a general base policy must have the capacity to contend not only with novel objectives but also with limited information, whether in the form of partial observability, uncertain dynamics, or unspecified goals. Thus, it must be effective across a spectrum of difficulty levels, delivering quick performance on easier tasks and sample-efficient adaptation over long timescales on complex ones.

With these desiderata in mind, we introduce a metric that guides goal generation using the agent's performance on goals after an adaptation budget. This contrasts with much prior work that evaluates goal desirability based on immediate performance without any task-specific adaptation (Sukhbaatar et al., 2017; Florensa et al., 2018). We pair this metric with an adversarially trained goal-generation system that proposes challenging yet achievable goals for the pre-trained policy to learn from. This yields an adaptive curriculum that maintains goals at intermediate difficulty and avoids uninformative extremes of too easy or unsolvable goals. Shifting toward a measure of difficulty that relies on post-adaptation performance aligns better with the intended evaluation setting, where the policy must face novel tasks that require adaptation. Moreover, it focuses training on more challenging tasks and leads to coverage of a broader distribution, with more tasks becoming easy and fewer remaining effectively unsolvable.

Anticipating the need for test-time adaptation naturally points to meta-learning (Duan et al., 2016; Finn et al., 2017), which explicitly optimizes for efficient learning. Meta-learning has primarily been studied with pre-training carried out over hand-designed task distributions, and its integration with unsupervised settings remains largely underexplored (Gupta et al., 2018). We bridge these directions by introducing an unsupervised meta-learning algorithm that meta-learns a base policy using an automatic curriculum of self-generated goals guided by our post-adaptation difficulty metric.

Our main contributions are as follows:

- We introduce a post-adaptation task-difficulty metric for guiding automatic goal generation and selection in unsupervised autotelic agents.

- We present ULEE, an unsupervised meta-learning method composed of (1) an in-context learner trained with self-imposed goals, (2) a difficulty-prediction network estimating post-adaptation performance, (3) an adversarial agent trained to find challenging candidate goals, and (4) a sampling strategy that selects goals within a desired difficulty range.

- We evaluate ULEE in complex yet computationally accessible environments across multiple timescales and types of generalization. It outperforms baselines and ablations in exploration, fast adaptation, fine-tuning to fixed environments, and fine-tuning in meta-learning settings. All code is open-sourced.

## 2 RELATED WORK

Several strategies have emerged to make use of reward-free objectives in RL. These include learning a world model (Sekar et al., 2020; Seo et al., 2022; Rajeswar et al., 2023), utilizing auxiliary objectives that aid representation learning (Jaderberg et al., 2016; Oord et al., 2018; Zhang et al., 2020a; Stooke et al., 2021; Schwarzer et al., 2021), and training agents with intrinsic rewards (Chentanez et al., 2004; Oudeyer et al., 2007; Schmidhuber, 2010) that lead to useful behaviors. The latter has been a major line of research into both pre-training and online exploration, with methods in both areas holding a close relationship. Most work in this line of research can be grouped into methods that consider predictions over some aspect of the environment and are guided by error, uncertainty, or progress in these predictions (Schmidhuber, 1991; Pathak et al., 2017; Burda et al., 2018; Pathak et al., 2019); those whose rewards come from measures of diversity in collected data (Bellemare et al., 2016; Hazan et al., 2019; Lee et al., 2019; Liu & Abbeel, 2021; Yarats et al., 2021); and those who seek to maximize empowerment (Klyubin et al., 2005; Mohamed & Jimenez Rezende, 2015; Gregor et al., 2016; Eysenbach et al., 2018; Sharma et al., 2019) or are guided by self-imposed goals (Sukhbaatar et al., 2017; Nair et al., 2018; Florensa et al., 2018; Pong et al., 2019; Forestier et al., 2022). Our work focuses on the latter and incorporates ideas from automatic curriculum learning (Bengio et al., 2009; Andrychowicz et al., 2017; Florensa et al., 2017; Matiisen et al., 2019; Jiang et al., 2021), which has proved capable of significantly accelerating learning in RL.

**Intermediate difficulty.** Previous work on unsupervised RL and automatic curriculum generation has explored dictating the agent's curriculum by continually selecting tasks of intermediate difficulty. In GoalGAN (Florensa et al., 2018), a GAN (Goodfellow et al., 2014) variant is trained to output goals that are neither too easy nor too hard for the agent. The goal space is fixed beforehand and remains the same for training and evaluation. The curriculum's aim is to accelerate learning across all feasible goals. Racaniere et al. (2019) address the same multitask setting but generate goals uniformly across all difficulty levels. Their goal generator is trained with custom objectives that promote goal validity, coverage, and controllable difficulty. They introduce a "judge" network trained on past goals to estimate success probability. Conditioning the judge and generator on environment observations enables them to handle procedurally generated and partially observable environments. Zhang et al. (2020b) propose a curriculum method that does not rely on adversarial training. They treat previously visited states as candidate goals and estimate goal difficulty from the epistemic uncertainty of the value function, approximated using an ensemble of value networks. They show that goals at both extremes of difficulty correspond to low uncertainty. AMIGo (Campero et al., 2020) is a method to improve online exploration in which a generative network is trained jointly with the agent to provide it with progressively more challenging yet achievable tasks. This interaction leads to the exploration of increasingly complex behaviors that assist the agent in finding the environment's sparse extrinsic reward. Like our work, these methods ground their curricula in difficulty-based metrics, though in settings that differ fundamentally from ours. None consider unsupervised pre-training followed by transfer to previously unseen tasks, and they don't consider scenarios where goal information isn't available. Instead, they train goal-conditioned policies and design curricula either for multitask learning or to accelerate progress on a single extrinsic task available alongside the unsupervised goals. Moreover, their difficulty estimates are derived from the agent's immediate performance, without allowing for task-specific adaptation. ASP (Sukhbaatar et al., 2017) is closer to our setup in that it also employs an adversarial policy ("Alice") to propose goals, but still optimizes a different objective. ASP is evaluated on a single external task using a goal-conditioned policy ("Bob"), which alternates between self-play and training on the target task. During self-play episodes, Bob is rewarded for either imitating or undoing Alice's actions. OpenAI et al. (2021) extend ASP with an imitation learning loss and scale it to evaluate the generalization of the goal-conditioned policy to held-out tasks. Beyond goal-based curricula, other works investigate adjusting task difficulty through the automatic selection or modification of training environments (Wang et al., 2019; Akkaya et al., 2019; Mehta et al., 2020; Team et al., 2021; 2023).

**Unsupervised meta-learning.** The setting of unsupervised meta-learning for RL, where meta-learning is performed on automatically acquired tasks, was first explored in Gupta et al. (2018). There, each latent skill $z$ discovered by Diversity is All You Need (DIAYN) (Eysenbach et al., 2018) is associated with a self-generated goal, and the discriminator probabilities $D(z|s)$ are used as the rewards for that goal. Jabri et al. (2019) extends this approach with a dynamic curriculum and evaluates it in partially observable visual environments. Alternatively, Mutti et al. (2022) learns a policy that maximizes the entropy of the induced state-visitation distribution, with particular emphasis on adverse environments.

**Ada.** Our work also bears similarities to Ada (Team et al., 2023). Ada meta-learns an in-context learning agent (Duan et al., 2016; Wang et al., 2016; Mishra et al., 2017) on a large task distribution using a curriculum and evaluates performance on novel tasks in a few-shot regime. Unlike our method, Ada trains solely on tasks produced by its environment generator and does not incorporate intrinsic goals. In addition, its policy is explicitly conditioned on goal and environment-dynamics information, and the learned policy is not evaluated as an initialization for extended fine-tuning.

## 3 METHOD

We introduce ULEE (Unsupervised Learning of Efficient Exploration), an unsupervised pre-training method in which an intrinsically motivated agent meta-learns an adaptive policy from a curriculum of self-imposed goals. This section outlines its main components. Pseudocode and additional details are provided in Appendices A.1 and A.2.

**Setting and Notation:** Both pre-training and evaluation are performed over families of partially observable Markov decision processes (POMDPs). We denote by $\mu^{\text{train}}$ and $\mu^{\text{eval}}$ the distributions of pre-training and evaluation environments, respectively. Agent interactions with each environment

$M$ occur in discrete time steps $t \in \{0, \ldots, T-1\}$, where $s_t \in \mathcal{S}$ is the underlying state of the environment at time step $t$, $o_t \sim \mathcal{O}(\cdot | s_t)$ is the observation the agent gets, and $a_t \in \mathcal{A}$ is the action it takes. Environments in our work differ in their initial state distribution $\rho_M$, transition dynamics $P_M(s_{t+1} | s_t, a)$, and reward function $r_M(s_t, a_t)$. In the pre-training phase, rewards are not provided by the environments but determined by the agent's own intrinsic goals. We write $\mu^{\text{unsup}}$ for the reward-free counterpart of $\mu^{\text{train}}$ and pair each $M \sim \mu^{\text{unsup}}$ with a goal-conditioned reward function to obtain full POMDPs. Although the policy operates on observations, for simplicity we give the goal-generation system and goal-conditioned reward function privileged access to full state information during pre-training.

## 3.1 PRE-TRAINED POLICY

The **Pre-trained Policy** $\pi$ is the component trained with self-imposed goals and the sole component evaluated at test time. Prior work on self-generated goals typically learns goal-conditioned policies. Although goal conditioning scales to broad goal distributions, it is ill-suited for evaluation settings in which (i) the goal to be achieved is unknown and must be discovered, (ii) there is no suitable way to encode the goal, or (iii) the goal representation is sufficiently out-of-distribution to be uninterpretable by the policy. These scenarios place goal-conditioned policies outside the regime where they excel – known goals with recognizable representations. Some methods address this by treating the goal-conditioned policy as a low-level controller within a hierarchy and deploying the high-level controller (Kulkarni et al., 2016; Vezhnevets et al., 2017; Sukhbaatar et al., 2018; Sharma et al., 2019). In contrast, we pre-train an *unconditioned* policy that can be directly deployed.

We investigate the agent's ability to solve goals under various adaptation timescales, noting that longer horizons better match the demands of challenging or unfamiliar tasks. To that end, we adopt a meta-learning approach to train $\pi$, which allows explicit optimization of exploration and adaptation over multiple episodes. In our implementation, we take a black-box approach (Duan et al., 2016; Wang et al., 2016), where the agent interacts with environments over a sequence of contiguous episodes and selects actions based on its full interaction history. By conditioning on past observations, actions, and rewards, the policy learns to adapt *in-context*. We refer to the whole multi-episode interaction as a *lifetime* and train $\pi$ to maximize its expected discounted lifetime return.

$$\mathcal{J}(\pi) = \mathbb{E}_{M \sim \mu^{\text{unsup}}, g \sim p(g|M)} \left[ \mathbb{E}_{\rho_M, P_M, \pi} \left[ \sum_{j=1}^{H} \sum_{t=0}^{T-1} \gamma^{(j-1)T+t} \, r_t^{(j)} \right] \right] \quad (1)$$

In Equation 1, $j \in \{1, \ldots, H\}$ indexes episodes within a lifetime, $r_t^{(j)}$ is the reward for the sampled goal $g$ at time step $t$ in episode $j$, $\gamma \in [0, 1)$ is the discount factor, and $p(g | M)$ is the evolving distribution of self-imposed goals induced by ULEE's goal-generation mechanism.

## 3.2 GOAL CURRICULUM

In our work, goals are obtained as functions of states via $f : \mathcal{S} \to \mathcal{G}$. An agent accomplishes goal $g$ at time step $t$ if $f(s_{t+1}) = g$, meaning the goal is reached with action $a_t$. Our method is flexible to the choice of $f$. Our procedure finds and selects goals that are of intermediate difficulty according to the current capabilities of the Pre-trained Policy. This prevents it from focusing on goals that are too hard to get any feedback on or too easy to require new capabilities, and thus provides a strong learning signal (Florensa et al., 2018; Zhang et al., 2020b). We define the difficulty of a goal $g$ for a policy $\pi$ in environment $M$ as the complement of $\pi$'s expected success rate over the last $K$ episodes of sequential interaction (Eq. 2). Thus, with $H$ being the total number of episodes in the lifetime, the performance over the first $H - K$ episodes, which the agent leverages to explore and adapt, is ignored.

$$d(g; \pi, M) \;=\; 1 - \mathbb{E}_{\rho_M, P_M, \pi} \left[ \frac{1}{K} \sum_{j=H-K+1}^{H} \mathbf{1}\Big\{ \exists t \in \{0, \ldots, T-1\} : f(s_{t+1}^{(j)}) = g \Big\} \right] \quad (2)$$

For each sampled environment $M \sim \mu^{\text{unsup}}$ and goal $g \sim p(g \mid M)$, running $\pi$ for one lifetime yields a single-sample empirical estimate of that task's difficulty $\tilde{d}(g; \pi, M)$. We implement the intermediate-difficulty curriculum with three subsystems that dynamically adjust $p(g \mid M)$.

### 3.2.1 GOAL PROPOSAL

A **Goal-search Policy** $\pi_{gs}$ is trained adversarially to reach hard goals. More specifically, it is trained to maximize $\mathcal{J}_{gs}(\pi_{gs}) = \mathbb{E}_{M, \pi_{gs}} \left[ \sum_{t=0}^{T-1} \gamma^t r_t^{gs} \right]$, where $\gamma$ is a discount factor and the goal-search rewards $r^{gs}$ are the difficulties of the candidate goals encountered (Eq. 3).

$$r_t^{gs} = r^{gs}(s_t; \pi, M) = d(f(s_t); \pi, M) \tag{3}$$

In each training environment $M$, the Goal-search Policy runs for $k$ episodes prior to the Pre-trained Policy. Let $s_0, s_1, \ldots$ denote the sequence of states visited across these episodes. Then, the multiset of candidate goals collected by $\pi_{gs}$ for $M$ is defined as $GC_M = \{f(s_t) : t \in \{0, n, 2n, \ldots\}\}$, where $n \in \mathbb{N}$ controls the temporal spacing between considered goals.

### 3.2.2 GOAL SELECTION

To decide with which goal to train the Pre-trained Policy among the candidate goals $GC_M$, we perform a random sample among goals at a desired level of difficulty. More specifically, we define a lower bound $LB$ and an upper bound $UB$ and sample uniformly among goals whose difficulty lies within these bounds (Eq. 4).

$$g_M \sim \text{Unif}(S), \quad S = \{g \in GC_M : LB \leq d(g; \pi, M) \leq UB\} \tag{4}$$

Here, $g_M$ is the goal that is sampled for environment $M$, and $\pi$ is the current Pre-trained Policy. If $S = \emptyset$, we instead sample uniformly from $GC_M$.

### 3.2.3 DIFFICULTY PREDICTION

In practice, there is no access to the goal difficulties needed to compute the goal-search rewards or to uncover which goals are in $S$. Even computing a single-sample empirical estimate requires running $\pi$ for multiple episodes, which becomes prohibitively expensive and sample-inefficient. Consequently, we introduce a **Difficulty Predictor** network that can estimate the difficulty of goals the agent has not faced. By replacing the true goal difficulties with the Difficulty Predictor estimates $\hat{d}$, we can approximate the behavior of Equations 3 and 4 without additional environment interactions.

As the difficulty of goals depends on the agent's evolving capabilities, we keep a buffer $B_g$ of triplets $(g, \xi_M, \tilde{d}(g))$ that holds only the most recent goals on which pre-training was performed. In each iteration, we train the Difficulty Predictor with a supervised L2 regression loss computed from goals in this buffer (Eq. 5).

$$\mathcal{L}_{\text{DP}}(\phi) = \frac{1}{|B_g|} \sum_{(g, \xi, \tilde{d}) \in B_g} \left( \hat{d}_\phi(g, \xi) - \tilde{d}(g) \right)^2. \tag{5}$$

The target $\tilde{d}(g)$ is a one-sample empirical estimate of goal difficulty obtained from the Pre-trained Policy's training lifetime on $g$. $\xi_M$ denotes a generic object (e.g., $s_0 \sim \rho_M$) that carries information regarding the environment $M$ on which the empirical estimate was obtained.

## 4 EXPERIMENTS

Unsupervised reinforcement learning methods can differ in the form of knowledge they aim to acquire, leading to various criteria for their evaluation. In this work, we assess the downstream utility of ULEE's Pre-trained Policy on novel tasks. Specifically, we ask:

Q1 What exploration capabilities does the Pre-trained Policy exhibit? (Sec. 4.3.1)

Q2 How well does the policy adapt with limited experience? (Sec. 4.3.2)

Q3 Does the Pre-trained Policy provide a strong initialization for fine-tuning under larger adaptation budgets? (Sec. 4.3.3)

Q4 Is the Pre-trained Policy an effective initialization for downstream meta-learning on curated task distributions? (Sec. 4.3.4)

Finally, Section 4.3.5 revisits Q1 and Q2 to test along an additional axis of generalization.

## 4.1 BENCHMARKS AND TASKS

We evaluate on distributions of pre-sampled, procedurally generated, partially observable grid environments from the JAX-based XLand-MiniGrid (Nikulin et al., 2024), which retains the diversity of XLand (Team et al., 2021) while enabling orders of magnitude faster experimentation. Environments contain a goal and a set of rules governing object-object and agent-object interactions (Fig. 1a). Various types of goals, rules, and objects exist. We use two distributions of 1 million pre-sampled combinations of {goal, rules, initial objects} adhering to the following specifications:

- **trivial:** no prerequisite rules must be triggered before achieving the goal; three distractor objects not involved in achieving the goal are present.
- **small:** 0-2 rules that the agent must trigger before the goal becomes achievable; 2 distractor objects and 0-2 distractor rules that lead to dead ends are present.

We allocate $100\,000$ combinations to $\mu^{\text{eval}}$ and the remaining 90% to $\mu^{\text{train}}/\mu^{\text{unsup}}$. All environments use a 13x13 grid, symbolic 5x5 observations, and an episode horizon of 256 steps with early termination upon success. Each distribution is instantiated with either four- or six-room layouts, yielding our three benchmarks: 4Rooms-Trivial, 4Rooms-Small, and 6Rooms-Small (Fig. 1).

The goal mapping $f : \mathcal{S} \rightarrow \mathcal{G}$ used during unsupervised pre-training encodes an inductive bias into which behaviors are worth mastering and thus affects the difficulty of generalizing to XLand-MiniGrid extrinsic goals. We consider a mapping $f_{\text{counts}}$ to the grid's per-object counts, where the agent succeeds when its interactions lead to the same count vector, and a mapping $f_{\text{grid}}$ that defines goals by the agent's position and the grid's exact configuration, excluding the agent's orientation and pocket; success requires reproducing that configuration. These mappings serve as reasonable task-agnostic proxies. We utilize $f_{\text{counts}}$ as the default mapping and evaluate $f_{\text{grid}}$ exclusively on 4Rooms-Small, making the distinction explicit. See Appendix A.2.1 for further details on the environments.

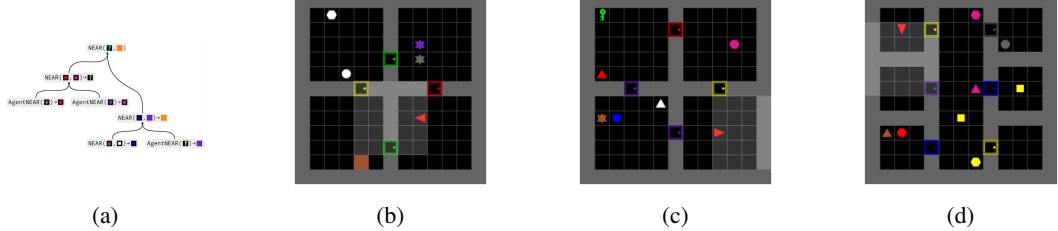

|     |     |     |     |
| --- | --- | --- | --- |
| (a) | (b) | (c) | (d) |

Figure 1: Panel (a), reproduced from Nikulin et al. (2024) (CC BY 4.0), shows a goal and the rules that must be triggered to achieve it, represented as a tree of depth 2. Analogously, tasks from the trivial and small benchmarks correspond to depth-0 and depth-1 trees. Panels (b)-(d) show example environments from the three benchmarks: 4Rooms-Trivial, 4Rooms-Small, and 6Rooms-Small.

## 4.2 BASELINES AND ABLATIONS

**Ablation Variants:** We study ablations of ULEE's goal-generation mechanism. A first set of ablations denotes each variant as *ULEE (goal_search + sampling)*. The goal_search component is *adversarial* when the Goal-search Policy is trained to reach difficult goals, and *random* when replaced by a random policy. The sampling component is *bounded* when goals are drawn uniformly from those of intermediate difficulty and *uniform* when drawn from all candidates. Our main method

(Sec. 3) corresponds to ULEE (adversarial + bounded), which we refer to simply as ULEE. In addition, we introduce ULEE (SED), which estimates difficulty from the Pre-trained Policy's immediate rather than post-adaptation performance. After meta-learning with tasks over multi-episode interactions, we compute their *single-episode difficulty* using $K$ additional episodes run as if they were first episodes (resetting memory before each). These extra episodes are excluded from the ablation step counts reported in figures and text.

**Baselines:** We compare our method against DIAYN (Eysenbach et al., 2018), an empowerment-based unsupervised RL method used as a pre-training baseline (Q1–Q3); PPO (Schulman et al., 2017) trained from scratch on fixed tasks as a standard model-free baseline (Q3); RND (Burda et al., 2018) trained from scratch on fixed tasks as a popular baseline that uses intrinsic rewards to improve online exploration (Q3); and RL$^2$ (Duan et al., 2016; Wang et al., 2016) trained from scratch on a curated task distribution as a standard meta-learning baseline (Q4).

**Implementations:** All policies are optimized with PPO and use the same Transformer-XL backbone (Dai et al., 2019; Parisotto et al., 2020), with separate MLP heads for the Actor and Critic. Variants with bounded sampling use $LB = 0.1$ and $UB = 0.9$. For DIAYN, the skill discriminator has access to the full state and conditions on $f(s)$. Further information is provided in Appendix A.2.

### 4.3 EXPERIMENTAL RESULTS

#### 4.3.1 EXPLORATION

To assess exploration capabilities, we measure the percentage of goals from $\mu^{\text{eval}}$ reached under budgets of 1-20 episodes. ULEE's Pre-trained Policy substantially outperforms random behavior (the common default when learning from scratch) and DIAYN pre-training, reaching more than twice as many goals at the 20-episode mark across all benchmarks (Fig. 2). Ablations highlight the need for mechanisms to avoid pre-training on trivial goals and show that guiding the curriculum by post-adaptation performance, rather than immediate success, becomes increasingly beneficial as benchmark difficulty grows. Variants using an adversarial Goal-search Policy achieve the best results, and bounded goal sampling proves effective when goal search is random.

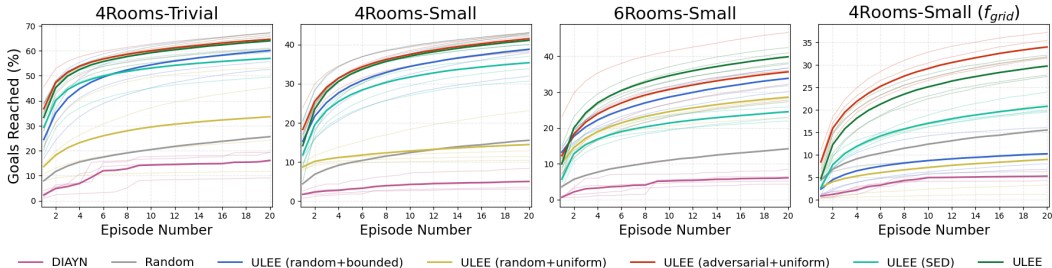

Figure 2: Percentage of $\mu^{\text{eval}}$ goals reached under different exploration budgets. A goal is considered reached at episode $j$ if it was achieved in any episode $\leq j$. Results are averaged across 4 seeds, with individual seeds overlaid as faint thin lines.

#### 4.3.2 FAST ADAPTATION

A policy must not only discover goal solutions but also know how to reliably achieve them once it does. Figure 3a evaluates gradient-free, few-shot adaptation over 30-episode lifetimes. ULEE's Pre-trained Policy leverages its interaction history to improve steadily, reaching up to a $3\times$ increase in mean return by the 30th episode. Variants trained with a random Goal-search Policy manage some adaptation early on but soon stagnate. For DIAYN, adaptation occurs at a single, discrete point by selecting the best-performing skill after all have been evaluated. Figure 3b reports the agent's post-adaptation return (measured as the mean over the last 10 episodes) by task percentile. ULEE outperforms all baselines and ablations. However, the hardest out-of-distribution tasks remain challenging. In two of the three benchmarks, no return was achieved on $60\%$ of tasks. Figure 3c shows that as the pre-training budget is increased (up to 5 billion steps), ULEE's post-adaptation performance continues to improve. DIAYN exhibits an initial gain followed by stagnation or even

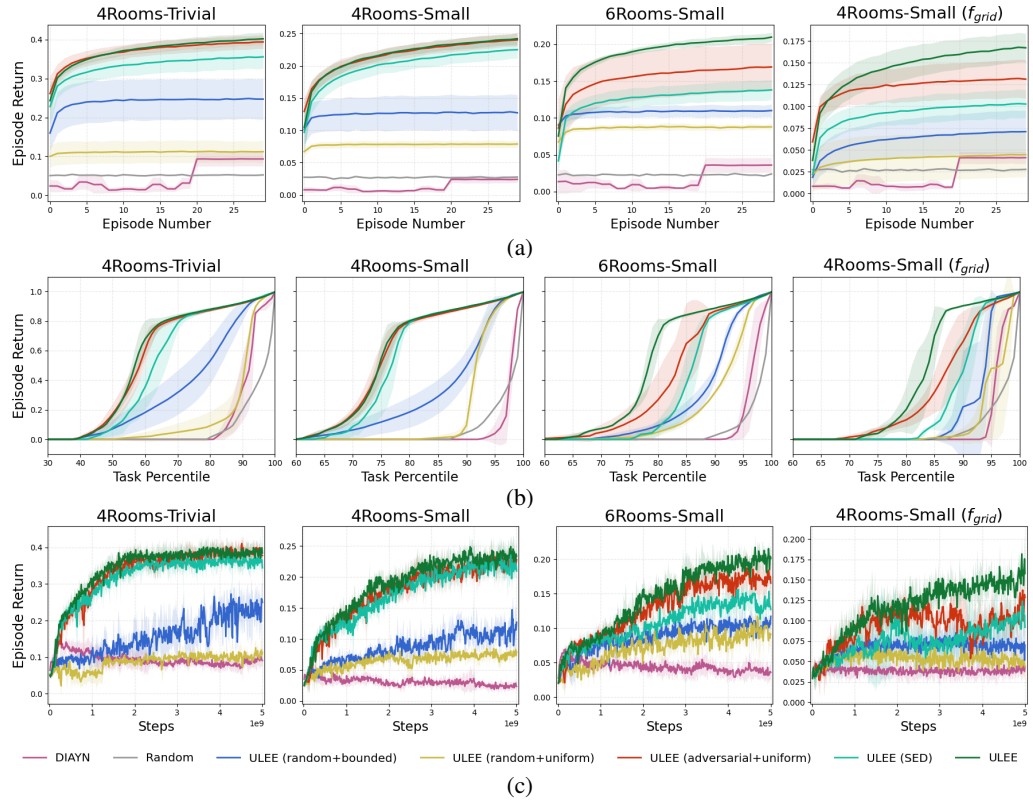

Figure 3: Evaluations on $\mu^{\text{eval}}$ tasks: (a) performance across episodes during task-specific adaptation, (b) few-shot performance by task percentile, (c) few-shot performance as pre-training on $\mu^{\text{unsup}}$ progresses. ULEE pre-training improves over baselines and ablations across all views. The legend in (c) applies to all panels. Reported steps for ULEE variants in (c) omit those from the Goal-search Policy, which adds 25%. Results are averaged over 4 seeds, with shaded regions indicating standard deviation.

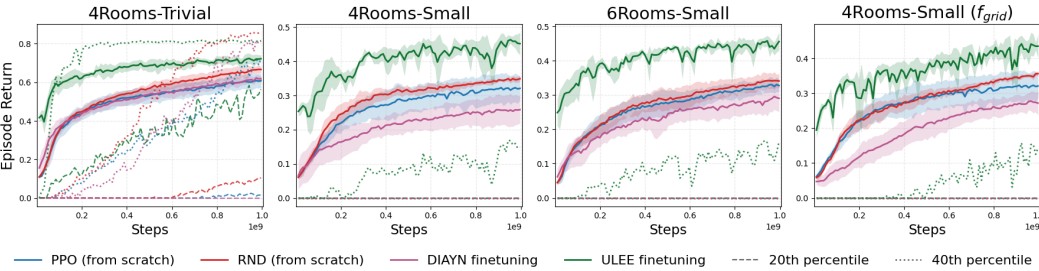

Figure 4: Mean, 40th, and 20th percentile returns on a fixed set of $\mu^{\text{eval}}$ tasks as learning on them progresses. Results are averaged over 4 seeds, with shaded regions indicating standard deviation.

decline as training continues. Lastly, across all views, using $f_{\text{counts}}$ over $f_{\text{grid}}$ during pre-training yields better results for ULEE, highlighting the value of $f$ as a source of inductive bias.

### 4.3.3 FINE-TUNING ON FIXED TASKS

Beyond few-shot performance, we evaluate the utility of pre-training when longer adaptation budgets are available. We sample 2 048 environments from $\mu^{\text{eval}}$ and, on this fixed set, compare training from scratch to fine-tuning policies pre-trained with ULEE or DIAYN. For budgets up to 1 billion steps, ULEE consistently outperforms the other methods in mean, 40th percentile, and 20th per-

centile returns (Fig. 4). While DIAYN provides a slight initial advantage over training from scratch in some benchmarks, this benefit is short-lived. The results in Figure 4 underscore the difficulty of the 4Rooms-Small and 6Rooms-Small benchmarks. Even when targeting a fixed, small subset of tasks with extrinsic rewards, no method manages to solve 80% of tasks, and without ULEE, at least 40% remain unsolved.

### 4.3.4 FINE-TUNING WITH SUPERVISED META-LEARNING

We next investigate whether ULEE provides a strong initialization for supervised meta-reinforcement learning. Figure 5 compares meta-learning from scratch to starting from ULEE's Pre-trained Policy. For budgets of up to 5 billion steps of meta-learning on $\mu^{\mathrm{train}}$, ULEE initialization yields higher mean, 40th percentile, and 20th percentile returns on $\mu^{\mathrm{eval}}$ tasks. In the 4Rooms-Small benchmark, using $f_{\mathrm{counts}}$ during the unsupervised phase leads to better results early in fine-tuning, but by 5 billion steps, the difference from $f_{\mathrm{grid}}$ is negligible.

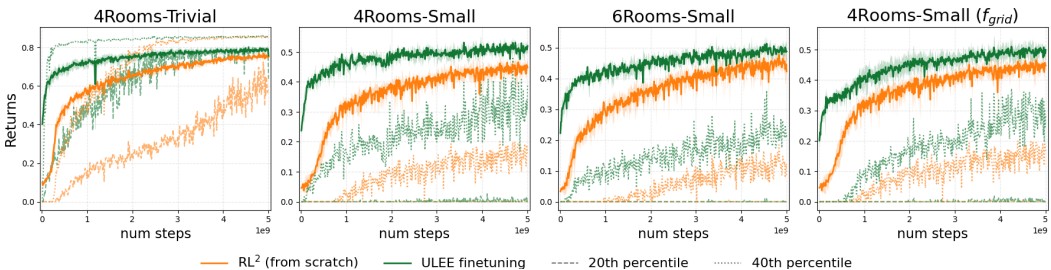

Figure 5: Mean, 40th, and 20th percentile returns on $\mu^{\mathrm{eval}}$ tasks as meta-learning on $\mu^{\mathrm{train}}$ progresses. Results are averaged over 4 seeds, with shaded regions indicating standard deviation.

### 4.3.5 GENERALIZATION TO NEW ENVIRONMENT STRUCTURES

Previous sections evaluated generalization to novel goals and dynamics, but the environments' layout (grid size and number of rooms) remained unchanged between pre-training and evaluation. To assess generalization along this additional axis, we test the performance of methods on a variety of classical MiniGrid tasks after pre-training on 4Rooms-Small (Table 1). In few-shot evaluations, ULEE ($f_{\mathrm{counts}}$) attains the best results, achieving a non-zero return on all environments. ULEE ($f_{\mathrm{grid}}$) and DIAYN ($f_{\mathrm{grid}}$) also improve substantially over a random policy.

Table 1: Mean return on MiniGrid tasks. ULEE and DIAYN methods were pre-trained on 4Rooms-Small and their performance is evaluated after 20 adaptation episodes. Results are averaged over 4 seeds, with standard deviations reported. The best-performing method is highlighted in bold.

| task | Random | DIAYN | DIAYN ($f_{\mathrm{grid}}$) | ULEE | ULEE ($f_{\mathrm{grid}}$) |
|---|---|---|---|---|---|
| BlockedUnlockPickUp | $0.02 \pm 0.00$ | $0.02 \pm 0.01$ | $0.03 \pm 0.00$ | $\mathbf{0.43 \pm 0.19}$ | $0.03 \pm 0.00$ |
| DoorKey-5x5 | $\mathbf{0.05 \pm 0.00}$ | $0.00 \pm 0.00$ | $\mathbf{0.08 \pm 0.14}$ | $0.27 \pm 0.35$ | $\mathbf{0.08 \pm 0.02}$ |
| DoorKey-8x8 | $\mathbf{0.01 \pm 0.00}$ | $0.00 \pm 0.00$ | $0.01 \pm 0.01$ | $0.24 \pm 0.24$ | $0.02 \pm 0.01$ |
| DoorKey-16x16 | $\mathbf{0.00 \pm 0.00}$ | $\mathbf{0.00 \pm 0.00}$ | $0.00 \pm 0.00$ | $0.11 \pm 0.12$ | $0.00 \pm 0.01$ |
| Empty-8x8 | $0.11 \pm 0.00$ | $0.18 \pm 0.19$ | $\mathbf{0.72 \pm 0.42}$ | $0.64 \pm 0.22$ | $0.70 \pm 0.25$ |
| Empty-16x16 | $0.05 \pm 0.00$ | $\mathbf{0.39 \pm 0.40}$ | $0.73 \pm 0.42$ | $0.23 \pm 0.15$ | $0.53 \pm 0.36$ |
| EmptyRandom-8x8 | $0.24 \pm 0.00$ | $0.18 \pm 0.23$ | $\mathbf{0.77 \pm 0.34}$ | $0.49 \pm 0.30$ | $0.61 \pm 0.33$ |
| EmptyRandom-16x16 | $0.15 \pm 0.00$ | $\mathbf{0.29 \pm 0.34}$ | $0.59 \pm 0.32$ | $0.17 \pm 0.10$ | $0.47 \pm 0.31$ |
| FourRooms | $0.02 \pm 0.00$ | $0.07 \pm 0.01$ | $\mathbf{0.13 \pm 0.02}$ | $0.14 \pm 0.03$ | $0.16 \pm 0.04$ |
| LockedRoom | $0.00 \pm 0.00$ | $0.00 \pm 0.00$ | $0.00 \pm 0.00$ | $\mathbf{0.01 \pm 0.00}$ | $0.00 \pm 0.00$ |
| MemoryS8 | $0.18 \pm 0.00$ | $0.09 \pm 0.05$ | $0.41 \pm 0.05$ | $0.47 \pm 0.10$ | $\mathbf{0.52 \pm 0.01}$ |
| MemoryS16 | $0.22 \pm 0.00$ | $\mathbf{0.31 \pm 0.20}$ | $0.42 \pm 0.18$ | $0.51 \pm 0.04$ | $0.51 \pm 0.06$ |
| Unlock | $0.03 \pm 0.00$ | $0.01 \pm 0.01$ | $0.10 \pm 0.05$ | $\mathbf{0.75 \pm 0.16}$ | $0.02 \pm 0.01$ |
| UnlockPickUp | $0.00 \pm 0.00$ | $0.00 \pm 0.00$ | $0.00 \pm 0.00$ | $\mathbf{0.68 \pm 0.15}$ | $0.00 \pm 0.00$ |

## 5 CONCLUSION

We present ULEE, an unsupervised meta-learning approach to induce a pre-trained policy with transferable capabilities. ULEE leverages an adversarial curriculum of self-imposed goals to train on tasks that are challenging but solvable for the policy under a given adaptation budget. After pre-training on reward-free grid worlds, we evaluate on novel, partially observable, procedurally generated environments. ULEE outperforms baselines and ablations in zero-shot and few-shot settings and provides a strong initialization for fine-tuning on both fixed tasks and curated meta-learning distributions. It demonstrates generalization to new goals, transition dynamics, and grid structures, and scales favorably with the size of the pre-training and adaptation budgets. Future work could refine or extend ULEE's components, for instance by introducing hierarchical structure into the meta-learned policy to address longer-horizon tasks. A complementary direction is to integrate vision-language models (VLMs) into the goal-proposal and reward-specification mechanisms to align pre-training with human-relevant tasks and potentially enhance applicability to real-world settings.

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

# A APPENDIX

## A.1 PSEUDOCODE

---

**Algorithm 1:** ULEE. Unsupervised Learning of Efficient Exploration

---

**1** **Initialize:** Pre-trained Policy $\pi$, Goal-search Policy $\pi_{gs}$, Difficulty Predictor $\hat{d}_\phi$, and goal buffer $B_g$
**2** **while** *training* **do**
**3**      Sample a batch of environments $\{M_i\}_{i=1}^N \sim \mu^{\text{unsup}}$
     ▷ Search for candidate goals
**4**      **foreach** $M_i$ **do**
**5**          Run $\pi_{gs}$; obtain candidate goals $GC_{M_i}$ (Sec. 3.2.1) and environment information $\xi_{M_i}$
**6**      **end**
     ▷ Sample goals of intermediate difficulty
**7**      **foreach** $M_i$ **do**
**8**          Compute predicted difficulties $\hat{d}_\phi(g, \xi_{M_i})$ for all $g \in GC_{M_i}$
**9**          Filter to target difficulty range: $S \leftarrow \{\, g \in GC_{M_i} : LB \leq \hat{d}_\phi(g, \xi_{M_i}) \leq UB \,\}$ (Eq. 4)
**10**          **if** $S \neq \emptyset$ **then**
**11**              Sample $g_{M_i} \sim \text{Unif}(S)$
**12**          **else**
**13**              Sample $g_{M_i} \sim \text{Unif}(GC_{M_i})$
**14**          **end**
**15**
**16**      **end**
     ▷ Train Pre-trained Policy
**17**      $D_{\text{goals\_successes}} \leftarrow \emptyset$
**18**      **while** *lifetimes not finished* **do**
**19**          $D_{\text{update}} \leftarrow \emptyset$
**20**          **foreach** $M_i$ **do**
**21**              advance $\pi$'s lifetime on $M_i$ under goal $g_{M_i}$; append transitions to $D_{\text{update}}$
**22**              check for success $\{\exists t \in \{0, \ldots, T-1\} : f(s_{t+1}) = g\}$ in completed episodes and update $D_{\text{goals\_successes}}$
**23**          **end**
**24**          update $\pi$ on $D_{\text{update}}$ to maximize meta-RL objective $\mathcal{J}(\pi)$ (Eq. 1)
**25**      **end**
**26**      Compute empirical difficulty estimates $\tilde{d}(g_{M_i})$ from $D_{\text{goals\_successes}}$ according to Eq. 2
**27**      Push $\{(g_{M_i}, \xi_{M_i}, \tilde{d}(g_{M_i}))\}_{i=1}^N$ into buffer $B_g$ (replace the $N$ oldest tuples if full)
     ▷ Train Difficulty Predictor
**28**      Update $\hat{d}_\phi$ on data from $B_g$ to minimize $\mathcal{L}_{\text{DP}}(\phi)$ (Eq. 5)
     ▷ Train Goal-search Policy
**29**      **for** *num\_gs\_updates* **do**
**30**          $D_{\text{update}} \leftarrow \emptyset$
**31**          **foreach** $M_i$ **do**
**32**              run $\pi_{gs}$; collect trajectories into $D_{\text{update}}$, setting $r_t^{gs} \leftarrow \hat{d}_\phi\big(f(s_t), \xi_{M_i}\big)$ (Eq. 3)
**33**          **end**
**34**          update $\pi_{gs}$ on $D_{\text{update}}$ to maximize $\mathcal{J}_{gs}(\pi_{gs}) = \mathbb{E}_{M, \pi_{gs}}\left[\sum_{t=0}^{T-1} \gamma^t r_t^{gs}\right]$
**35**      **end**
**36** **end**

---

## A.2 EXPERIMENTAL DETAILS

### A.2.1 ENVIRONMENTS AND GOAL MAPPING

Our benchmarks are derived from XLand-MiniGrid. This section extends the information given in Section 4.1; for full details, see Nikulin et al. (2024). Each environment offers six discrete primitive actions. There are 70 unique objects that can appear in the environments. Objects are identified by type (e.g., *key*, *ball*, *pyramid*) and color. The (extrinsic) task space spans 13 goal types (e.g., `AgentHoldGoal(a)` – whether agent holds a; `TileOnPositionGoal(a, x, y)` – whether $a$ is on $(x,y)$ position; `TileNearLeftGoal(a, b)` – whether $b$ is one tile to the left

of $a$). The rule space spans 10 rule types (e.g., `AgentHoldRule(a)` $\rightarrow$ `c` – If agent holds $a$ replaces it with $c$; `TileNearRightRule(a, b)` $\rightarrow$ `c` – If $b$ is one tile to the right of $a$, replaces one with $c$ and removes the other). Each grid cell is associated with a corresponding ID that identifies its kind (e.g., wall, red pyramid, empty space). $f_{\text{counts}}$ maps states to a vector that records the number of occurrences of each ID. Thus, an intrinsic goal under $f_{\text{counts}}$ specifies how many instances of each unique object should appear on the grid. For both intrinsic and extrinsic tasks, a reward is given only upon success. If the agent succeeds in time step $t$, it receives a reward of $1.0 - 0.9 \cdot (t/\text{max\_steps})$, where max_steps is the maximum number of steps the agent can take before the episode is considered a failure. We use early termination upon success. The agent's $5 \times 5$ observations are not occluded by walls.

Our configuration for benchmarks differs from XLand-MiniGrid's default behavior in two ways: (i) for 13x13 grids, we set the episode horizon (max_steps) to 256 rather than the default 509; (ii) we sample the initial state $s_0 \sim \rho_M$ once at the beginning of each lifetime and keep it fixed across all episodes within that lifetime. The default configuration resamples $s_0$ at the start of every episode.

Our evaluations on MiniGrid tasks in Section 4.3.5 use the default configurations of XLand-MiniGrid's pre-built MiniGrid environments.

### A.2.2 HARDWARE AND SEEDS

Each algorithm is trained with 4 random seeds. For every training seed, we draw two additional independent seeds: one to construct the benchmark split between $\mu^{\text{train}}/\mu^{\text{unsup}}$ and $\mu^{\text{eval}}$, and one to run the fine-tuning experiments (Sections 4.3.3 and 4.3.4). Hyperparameters for each method are held fixed across benchmarks and seeds. All experiments are executed on a single NVIDIA RTX 4090 GPU.

### A.2.3 ALGORITHM IMPLEMENTATIONS AND HYPERPARAMETERS

**Architectures**

**Policies:** Policies use a Gated Transformer-XL architecture with 2 blocks, hidden size 192, and 4 attention heads (head dimension 48). Gates use a bias of 2. The network attends to cached activations of up to 128 steps into the past, yielding an effective memory horizon of 256 (memory_len $\times$ num_blocks). During training, gradients are propagated for up to 64 steps. Two separate MLP heads, each with a size of $(256, 256)$, are used as the Actor and Critic. Each full policy network has $\approx 1.7M$ parameters. ReLU activations are used throughout. For meta-learned policies (ULEE's Pre-trained Policy and the RL[2] baseline), the per-step input is $\{o_t, d_t, a_{t-1}, r_{t-1}\}$, where $d_t \in \{0, 1\}$ indicates the beginning of a new episode (dummy $a_{t-1}$ and $r_{t-1}$ are used at the start of each lifetime). The meta-learner state is reset only upon encountering a new environment (at the start of a lifetime).

**Difficulty Predictor and discriminator:** ULEE's Difficulty Predictor and DIAYN's discriminator are MLPs of size $(256, 256)$ that operate on learned encodings of their inputs (see *Input encodings* below). Both have access to full state information during pre-training. DIAYN's discriminator predicts latent skills from encodings of visited states $q(z|f(s))$, where $f$ coincides with ULEE's goal-mapping function and serves as a source of inductive bias. The Difficulty Predictor's inputs are $f_{\text{grid}}(s_g)$ (with $f(s_g) = g$) and $\xi_M = f_{\text{grid}}(s_0)$, where $s_0 \sim \rho_M$ (see 3.2.3). $f_{\text{grid}}$ removes the agent's orientation and pocket information from the state data.

**RND target and predictor:** RND's target and predictor are MLPs with two hidden layers of size 256. They operate on encodings of observations. For benchmarks using $f_{\text{counts}}$, each observation is mapped to a vector that records the number of occurrences of each cell type; the MLP is applied directly to this vector and produces a 256-dimensional output. For benchmarks using $f_{\text{grid}}$, observations are first encoded with a small CNN (see *Input encodings* below); the MLP is then applied to this encoding with an output dimensionality of 64. RND is used as a baseline in Section 4.3.3, where agents are trained to solve $2\,048$ extrinsic tasks. Since a single policy is trained across all tasks, we also use a single shared target-predictor pair.

**Input encodings:** To process the symbolic grid observations and states, each cell's symbol is embedded via two learned embedding tables (shape and color). Then, a convolutional neural network

is applied, with the final output flattened. Additional inputs, when present, are concatenated to the flattened vector. Actions and skills have their own embedding tables. All learned embedding tables produce 16-dimensional vectors. For the Difficulty Predictor, the two input states are concatenated across the channel dimension before applying the CNN.

**DIAYN skills:** DIAYN is trained with 10 skills (following Gaya et al. (2021) and Kayal et al. (2025)). We add a fixed per-skill bias vector to the policy logits; these vectors are orthogonally initialized and scaled by a factor of 8. We find this addition important for learning distinguishable skills.

### Training

The pre-training phase for ULEE and DIAYN spans 5 billion environment steps. We collect experience in parallel from batches of $2\,048$ environments and, after $5\,120$ steps per environment, resample a new batch from $\mu^{\text{unsup}}$. ULEE updates its policy $\pi$ every 256 steps per environment, while DIAYN updates every 512 steps. Policy updates for all algorithms use PPO with the Adam optimizer (hyperparameters in Table 2).

For ULEE, an additional $25\%$ of steps (on top of the counts reported above) are allocated to the Goal-search Policy $\pi_{gs}$. For each newly sampled environment $M$, $\pi_{gs}$ first executes two 256-step episodes to collect candidate goals. It adds one candidate to $GC_M$ for every 15 states visited (Section 3.2.1). After the Pre-trained Policy and Difficulty Predictor are updated with data from $M$, $\pi_{gs}$ executes three additional 256-step episodes during which it is trained to reach difficult goals (Section 3.2.1).

Task difficulty (Equation 2) is estimated empirically as the Pre-trained Policy's success rate over its last five episodes on that task. For bounded sampling, we use a difficulty lower bound $LB$ of $0.1$ and upper bound $UB$ of $0.9$. The goal buffer $B_g$ used to train the Difficulty Predictor stores goals (together with their empirical difficulty estimates) from the last five batches of $2\,048$ intrinsic goals ($|B_g| = 5 \times 2048$). After the Pre-trained Policy $\pi$ finishes training on a new batch and the corresponding goals are added to $B_g$, we train the Difficulty Predictor for 2 epochs over the entire buffer using a minibatch size of 256 and a learning rate of $1 \times 10^{-4}$ (Sec. 3.2.3).

DIAYN's discriminator is updated every 512 steps per environment and is trained for a single epoch over all collected transitions with a learning rate of $3 \times 10^{-4}$.

For RND, we compute advantages for extrinsic and intrinsic rewards using the same $\gamma$ and $\lambda$ values as in Table 2. These are combined using an extrinsic-reward coefficient of $1$ and an intrinsic-reward coefficient of $0.1$ before policy updates. The predictor network is updated every 256 steps per environment and trained for a single epoch over all collected transitions. Its learning rate is $1 \times 10^{-4}$ for benchmarks that use $f_{\text{counts}}$ and $1 \times 10^{-5}$ for benchmarks that use $f_{\text{grid}}$.

### A.2.4 EXPERIMENTAL PROCEDURES

**Exploration:** For each seed in Figure 2, results are obtained from evaluation on a set of $16\,384$ environments sampled from $\mu^{\text{eval}}$. Methods interact for 20 sequential episodes with each environment. A goal is considered reached at episode $j$ if it was achieved in any episode $\leq j$. For DIAYN, we run two episodes with each skill, randomizing their order. Each skill is used at least once before any repeat.

**Fast adaptation:** For each seed in Figures 3a and 3b, results are obtained from evaluation on a set of $16\,384$ environments sampled from $\mu^{\text{eval}}$. Methods interact for 30 sequential episodes with each environment. For DIAYN, we run two episodes conditioning on each skill, followed by ten episodes with the best-performing skill. Figure 3a shows results across all 30 episodes, whereas Figure 3b reports an aggregate performance over the last 10. Figure 3c tracks performance on $\mu^{\text{eval}}$ throughout pre-training on $\mu^{\text{unsup}}$; at each evaluation point, we sample a fresh set of $1\,024$ environments from $\mu^{\text{eval}}$ and measure the mean performance over the last 5 episodes of 25-episode interactions.

**Fine-tuning on fixed tasks:** For Section 4.3.3, each seed is fine-tuned on a fixed set of $2\,048$ environments sampled from $\mu^{\text{eval}}$. Figure 4 tracks performance on this fixed set throughout training, with performance measured as the mean over the last 10 episodes of 30-episode interactions. For DIAYN, before fine-tuning we run 10 episodes of each skill per environment. In each environment,

Table 2: Hyperparameters for policy updates. All policies are updated using PPO with the Adam optimizer. Entropy coefficients are listed separately as they vary across different methods.

| Hyperparameter | Value |
|---|---|
| Learning rate | $2 \times 10^{-4}$ |
| Adam $\epsilon$ | $1 \times 10^{-5}$ |
| Discount factor $\gamma$ | 0.99 |
| Update epochs | 1 |
| Number of minibatches | 16 |
| Clipping $\epsilon$ | 0.2 |
| GAE $\lambda$ | 0.95 |
| Value loss coefficient | 0.5 |
| Max gradient norm | 0.5 |
| **Entropy coefficient:** | |
| Goal-search Policy | 0.01 |
| Pre-trained Policy | 0.005 |
| DIAYN | 0.03 |
| DIAYN when fine-tuning | 0.01 |
| PPO (from scratch) | 0.005 |
| RND | 0.005 |
| $RL^2$ (from scratch) | 0.005 |

we select the best-performing skill and keep it fixed during fine-tuning. These preparatory steps are excluded from the figure.

**Fine-tuning with supervised meta-learning:** Figure 5 reports performance on $\mu^{\text{eval}}$ throughout meta-learning on $\mu^{\text{train}}$. At each evaluation point, we sample a fresh set of 1 024 environments from $\mu^{\text{eval}}$ and measure the mean performance over the last 5 episodes of 25-episode interactions.

**MiniGrid:** In Section 4.3.5, results for each seed on each MiniGrid environment are averaged over 2 048 runs; each run consists of 30 sequential episodes, and we report the performance over the last 10 episodes.

### A.2.5 Hyperparameter Selection and Stability

As is common in meta-learning and unsupervised pre-training, conducting an exhaustive hyperparameter search was infeasible due to the length and computational cost of each run. Our hyperparameter studies therefore aimed to: (1) assess whether ULEE requires careful tuning in order to learn, and (2) identify a reasonable, stable configuration rather than optimize for the best possible values. Given this constrained scope, we do not present quantitative comparisons or prescriptive recommendations on the choice of hyperparameters. Instead, this section documents the explorations that were performed and the observations that informed our final choices.

Our investigations found that ULEE does not require careful tuning to learn effectively. Across all configurations we tried, we observed no signs of collapse, and the results reported in the main paper were obtained after limited tuning. All tuning and stability-analysis experiments were run on seeds that were different from those used in the final experimental results reported in Section 4.

When selecting hyperparameters for each method, we focused mainly on the following criteria:

- DIAYN: We monitored performance on tasks sampled from $\mu^{\text{train}}$ throughout unsupervised pre-training on reward-free environments from $\mu^{\text{unsup}}$. We additionally inspected skill distinguishability (via per-skill heatmaps on empty environments and via the discriminator loss) and the entropy of each skill. When searching for fine-tuning hyperparameters (Section 4.3.3), we evaluated performance on fixed extrinsic-reward tasks as fine-tuning progressed.

- $RL^2$ (from scratch): We monitored performance on tasks sampled from $\mu^{\text{train}}$ during supervised meta-learning on tasks from $\mu^{\text{train}}$.

- PPO (from scratch): We monitored performance on a set of $2\,048$ tasks sampled from $\mu^{\text{train}}$ while training on them.

- RND: We monitored performance on a set of $2\,048$ tasks sampled from $\mu^{\text{train}}$ while training on them.

- ULEE: We monitored performance on tasks sampled from $\mu^{\text{train}}$ throughout unsupervised pre-training on reward-free environments from $\mu^{\text{unsup}}$. Importantly, during the selection process, we did *not* inspect any of the evaluation metrics presented in Figures 2, 3, 4, and 5. In particular, we did not test or select hyperparameters based on downstream fine-tuning performance. We also did not tune any hyperparameters specific to fine-tuning policies pre-trained with ULEE; for those experiments we reused the values employed during pre-training.

Below we summarize which hyperparameters were varied and which were kept fixed. For those that were varied, our tuning process was limited in several ways: at most three hyperparameters were varied at a time, each combination was tested with only 1–2 seeds, experiments were run exclusively on the 4Rooms-Small benchmark, and runs were shorter than those used in the final results.

**Network architectures:** We did not perform any hyperparameter tuning on network architectures.

**Number of steps and parallel environments:** We did not tune the number of parallel environments, the split of {goal, rules, initial objects} tuples between $\mu^{\text{train}}$ and $\mu^{\text{eval}}$, the maximum number of steps per episode, the number of steps per environment before resampling, or the total number of training steps.

**Policy updates:** We tested variations of the entropy coefficient, discount factor $\gamma$, and learning rate. All other hyperparameters were fixed in advance. The values explored were:

- DIAYN: entropy coefficient $\{0.003, 0.005, 0.01, 0.02, 0.03, 0.05, 0.1\}$, learning rate $\{3 \times 10^{-5}, 1 \times 10^{-4}, 2 \times 10^{-4}, 3 \times 10^{-4}, 5 \times 10^{-4}\}$.

- DIAYN (fine-tuning): entropy coefficient $\{0.005, 0.01, 0.03\}$, learning rate $\{2 \times 10^{-4}\}$.

- RL$^2$: entropy coefficient $\{0.001, 0.003, 0.005, 0.007, 0.01, 0.03, 0.05\}$, learning rate $\{1 \times 10^{-4}, 2 \times 10^{-4}, 3 \times 10^{-4}, 5 \times 10^{-4}, 1 \times 10^{-3}\}$, discount factor $\{0.99, 0.995, 0.999\}$.

- PPO: entropy coefficient $\{0.003, 0.005, 0.01\}$, learning rate $\{3 \times 10^{-5}, 1 \times 10^{-4}, 2 \times 10^{-4}, 3 \times 10^{-4}, 5 \times 10^{-4}\}$.

- RND: entropy coefficient $\{0.003, 0.005, 0.01\}$, learning rate $\{3 \times 10^{-5}, 2 \times 10^{-4}, 5 \times 10^{-4}\}$.

- ULEE Pre-trained Policy: entropy coefficient $\{0.003, 0.005, 0.01, 0.03\}$, learning rate $\{3 \times 10^{-5}, 1 \times 10^{-4}, 2 \times 10^{-4}, 3 \times 10^{-4}, 5 \times 10^{-4}\}$.

- ULEE Goal-search Policy: entropy coefficient $\{0.003, 0.005, 0.01, 0.03\}$, learning rate $\{2 \times 10^{-4}\}$.

We found that learning rates between $1 \times 10^{-4}$ and $3 \times 10^{-4}$ worked well across all methods. Using discount factors of $0.995$ and $0.999$ in RL$^2$ strongly hindered performance.

**DIAYN-specific hyperparameters:** We varied the dimensionality of skill embeddings $\{16, 128\}$, the discriminator learning rate $\{3 \times 10^{-5}, 1 \times 10^{-4}, 3 \times 10^{-4}\}$, and the number of steps per environment per update $\{256, 512, 1024\}$. We also experimented with adding a fixed per-skill bias vector to the policy logits (Appendix A.2.3). This proved important for obtaining distinguishable skills in setups where the agent's and objects' initial positions are randomized. We explored two initializations for the bias vectors {orthogonal, random directions in the unit sphere} and several scale factors $\{1, 5, 8, 10, 15, 20\}$. Both initialization schemes were effective and scale factors between 5 and 10 performed best, though smaller and larger values still helped. Additionally, we tested disabling the policy updates for the first 10 batches of environments, which did not help, and varying the initialization of the discriminator final layer, which improved stability.

**RND-specific hyperparameters:** We varied the dimensionality of the embedding produced by the target network $\{32, 64, 256\}$, the intrinsic-reward coefficient $\{0.02, 0.05, 0.1, 0.3, 0.5, 1, 2\}$, and the predictor-network learning rate $\{1 \times 10^{-5}, 5 \times 10^{-5}, 1 \times 10^{-4}, 3 \times 10^{-4}\}$.

**ULEE-specific hyperparameters:**

*Difficulty Predictor.* We varied its learning rate $\{3 \times 10^{-5}, 1 \times 10^{-4}, 3 \times 10^{-4}, 5 \times 10^{-4}\}$ and the number of batches of intrinsic goals stored in the training buffer $\{1, 3, 5\}$. ULEE learned under all configurations. Differences were generally modest; a learning rate of $5 \times 10^{-4}$ produced the largest performance drop.

*Goal-search Policy and entropies.* As noted earlier, we varied the entropy coefficient of the Goal-search Policy and Pre-trained Policy updates. For the Goal-search Policy, we observed little effect on performance. For the Pre-trained Policy, the different values influenced the shape of pre-training curves but did not reveal a clear best choice. We also varied the number of Goal-search Policy updates per batch of environments (`num_gs_updates` in Algorithm 1) $\{3, 6\}$ and observed similar results.

*Goal sampling strategy.* We tested three sampling schemes: (1) uniform sampling from goals within difficulty bounds (Section 3.2.2), with bounds $\{(0.1, 0.9), (0.3, 0.7)\}$; (2) sampling a target difficulty for each environment $M$ from a Gaussian with mean in $\{0.4, 0.6, 0.8\}$ and standard deviation in $\{0.2, 0.4\}$ and picking the goal from $GC_M$ whose predicted difficulty is closer to that value; (3) assigning weights to candidate goals in $GC_M$ (based on their estimated difficulty) using a Gaussian density function with mean in $\{0.5, 0.6\}$ and standard deviation in $\{0.2, 0.4\}$ and sampling in accordance with these weights. ULEE learned under all schemes and hyperparameters tried. The best performance was obtained with scheme 2 and 3 using a Gaussian mean of $0.6$ and standard deviation of $0.2$, and with scheme 1 using a lower bound $LB$ of $0.1$ and upper bound $UB$ of $0.9$.

*Steps per update.* We varied the number of steps per environment per update $\{256, 512\}$ and observed faster learning with the smaller value.

No search was performed over hyperparameters not mentioned in this section. As noted earlier, our hyperparameter search was limited in scope: the goal was to identify reasonable values and evaluate whether ULEE exhibited failure modes or strong sensitivity to particular hyperparameters. Thus, the final set of hyperparameters should not be interpreted as either optimal or deeply tuned. Ablations used the same hyperparameters selected for ULEE.

## A.3 LEARNING PROGRESS

A complementary line of work in automatic curriculum generation employs *learning progress* (LP), rather than difficulty, as the guiding metric for goal selection (Baranes & Oudeyer, 2013; Colas et al., 2019; Forestier et al., 2022; Kovač et al., 2022). Learning progress is estimated by tracking an agent's performance on tasks over time and quantifying its rate of change. Tasks exhibiting the largest improvements (or deteriorations) are prioritized. Typically, LP is computed from changes in the policy's *immediate* performance as training proceeds, without allowing for any task-specific adaptation. Analogous to difficulty-based methods, LP can be adapted so as to measure changes in *post-adaptation* performance. Let $\tau$ denote training time and $\pi_\tau$ the policy after training for $\tau$. If $\tilde{d}(g; \pi, M)$ denotes a post-adaptation performance metric on task $(g, M)$, e.g., an empirical estimate of Eq. 2, then a possible LP-style goal desirability score is $LP_{\text{post}}(g, M; \tau) = |\tilde{d}(g; \pi_\tau, M) - \tilde{d}(g; \pi_{\tau - \Delta\tau}, M)|$, where $\Delta\tau > 0$ is the measurement window. Investigating post-adaptation learning progress and its interplay with post-adaptation difficulty-based curricula and unsupervised goal generation is a promising direction for future work.

## A.4 USE OF LARGE LANGUAGE MODELS

Large language models (LLMs) were used solely as coding assistants and for polishing the paper's writing. They were not used to generate, develop, or analyze the underlying scientific content or technical contributions.

