# OpenReview forum: "Unsupervised Learning of Efficient Exploration: Pre-training Adaptive Policies via Self-Imposed Goals"
_ICLR.cc/2026/Conference — ICLR 2026 Poster_

### Official Review · Reviewer_QuSP · 2025-10-15

**Soundness:** 3
**Presentation:** 4
**Contribution:** 3
**Rating:** 8
**Confidence:** 5

**Summary:**

The paper tackles meta‑reinforcement learning by combining in‑context RL with a curriculum for self‑generated goals. An ICRL agent is pre‑trained to reach goals proposed by a curriculum pipeline. This pipeline first optimizes a goal‑search policy to estimate the agent’s current capability, then a selector chooses goals within calibrated success bounds. On XLand‑MiniGrid benchmarks, the proposed method, ULEE, improves exploration and adaptation, generalizing to novel objectives, dynamics, and map structures, and achieves better zero‑shot and few‑shot performance.

**Strengths:**

- *Intuitive formulation:* ULEE frames meta-RL as in-context adaptation guided by self-imposed goals, yielding a conceptually clean pipeline (goal search → capability estimation → goal selection). This makes the approach easy to reason about and implement.
- *Solid empirical evidence:* On XLand–MiniGrid, ULEE consistently improves zero-shot and few-shot performance over strong baselines (learning from scratch, DIAYN, alternative curricula), with generalization to new objectives, dynamics, and map structures.
-  *High-quality writing and presentation:* The paper is clearly written, with strong contextualization relative to prior work and a well-structured narrative that explains the curriculum components (goal search, capability estimation, selector) and their interplay, which aids reproducibility and understanding.

**Weaknesses:**

- Despite the intuitive design, the pipeline introduces multiple components and hyperparameters (e.g., difficulty/success bounds), raising concerns about training stability and sensitivity.
- With adversarial goal-generation, does the search ever propose “degenerate” goals (e.g., trivially satisfiable or reward-hacking)? Although the goal selector may mitigate this issue, randomness in sampling can lead to training collapse.

**Questions:**

- Why not integrate the selector’s criterion directly into the goal-search reward—for example, use a $|x−0.5|$-shaped reward over success probability or estimated difficulty to penalize goals that are too hard or too easy? What are the trade-offs?
- Could classical exploration designs like UCB (or Thompson sampling) be used to guide goal sampling—treating difficulty bands or goal families as arms—and does this improve sample efficiency or stability compared to the current selector?

---

> ### Author Response · Authors · 2025-11-20
>
> Thank you for your time, detailed comments, and for your positive assessment of the introduced approach and empirical results. We respond to your concerns and questions below.
>
> > W1. *"Despite the intuitive design, the pipeline introduces multiple components and hyperparameters (e.g., difficulty/success bounds), raising concerns about training stability and sensitivity."*
>
> Thank you for raising this concern regarding the training stability and sensitivity of ULEE. We have addressed this in our general comment and in the newly added appendix section *“Hyperparameter Selection and Stability”*, which documents the hyperparameters explored and the observed stability across the configurations we tried.
>
>
> > W2. *"With adversarial goal-generation, does the search ever propose “degenerate” goals (e.g., trivially satisfiable or reward-hacking)? ... ."*
>
> We do not view consistently proposing easy goals as a likely failure mode, except in the case where discovering difficult goals is itself too hard. The Goal-search Policy is motivated to avoid easy goals, and any that it does visit are filtered in the goal selection/sampling step. Trivial goals could be selected if the Difficulty Predictor misclassifies them as intermediate or hard. However, once the Pre-trained Policy solves such goals, the predictor is trained with the correct label. As a result, we do not expect risk of persistent misclassification of trivial goals as being of intermediate or hard difficulty (as long as the Difficulty Predictor is a strong enough learner).
>
> We recognize the possibility of the Goal-search Policy ‘reward hacking’ in environments where it can reach very hard goals without passing through goals of intermediate difficulty. In that case, it could collect rollouts with high rewards but no useful goals to train on. However, we can’t identify typical environments where such behaviour would naturally occur. Importantly, any proposed goal is by construction reachable by the Pre-trained Policy, since the Goal-search Policy had to reach it in order to propose it.
>
> Moreover, if the goal-generation system exhibits unhelpful behavior in a particular environment, its impact is: (1) diluted by the contribution from the other 2047 environments in the batch, and (2) short lived, since each environment is only trained on once and unlikely to be revisited.
>
> Finally, in the new appendix section *“Hyperparameter Selection and Stability”* (mentioned in our general response), we report that we did not observe training collapse under any of the configurations we tried

---

> > ### Author Response · Authors · 2025-11-20
> >
> > > Q1. *"Why not integrate the selector’s criterion directly into the goal-search reward. ... ."*
> >
> > During development, we indeed considered rewarding the Goal-search Policy directly for intermediate-difficulty goals, since these are the ones we ultimately target in the selection step. However, we decided to keep the current objective that rewards higher difficulty for three main reasons:
> >
> > - We assume that reaching very difficult goals requires passing through intermediate-difficulty goals. Given that the Goal-search Policy starts its rollouts from trivially satisfiable goals, we view this assumption as reasonable for the studied and most other environments. Under a $|x-0.5|$ style reward, once the policy reaches a goal of intermediate difficulty, it has no incentive to keep exploring further. In contrast, when using difficulty as the reward, the policy remains motivated to search for even harder goals. In doing so, it can encounter new goals of intermediate difficulty, and increase the overall diversity of intermediate-difficulty goals found. Since we end up using one goal per environment, the fraction of time spent on goals of intermediate difficulty does not matter, but having a more diverse set of candidates might.
> >
> > - By learning to target difficult goals directly, the Goal-search Policy implicitly learns to reach a pool of future intermediate difficulty goals, as many current hard goals become accessible as the Pre-trained Policy improves. If instead, the Goal-search Policy were trained only to reach intermediate difficulty goals, these goals would tend to become easy over time, and rollouts could then contain no intermediate-difficulty goals. It would then require additional training for the Goal-search Policy to learn to reach new regions of intermediate difficulty.
> >
> > - Under a $|x-0.5|$ style reward, a bad rollout of the Goal-search Policy could involve it reaching a goal it believes to be of intermediate difficulty and then staying there, even if the goal is in fact easy. While the policy could also mistake an easy goal for a hard one, we consider this less likely in practice.
> >
> > While we found these considerations convincing when choosing among the two options, we did not ablate the decision experimentally. We view the choice of objective optimized by a second interacting agent as an interesting research direction that could lead to improvements. However, we leave this for future work.
> >
> >
> > > Q2. *"Could classical exploration designs like UCB (or Thompson sampling) be used to guide goal sampling ... ."*
> >
> > It is correct that our goal sampling mechanism is static, and only which goals lie within the target difficulty range evolves as training progresses. It would be interesting to see whether treating difficulty bands as arms in a bandit setup helps find the most useful difficulty range automatically and find whether it should change during training. To the best of our knowledge there are no works that explore such an evolving difficulty range.
> >
> > From an implementation perspective, however, it is not obvious what the bandit should optimize during unsupervised pre-training, as extrinsic rewards are not available. Optimizing for performance on the sampled intrinsic goals would drive the bandit toward trivially easy goals. A more appropriate signal could be based on something like learning progress or improvement on an independently sampled evaluation set of intrinsic goals. These signals however, depart from the classic bandit setting, as the distribution of rewards would change as training progresses. Regarding using ‘goal families’ as arms, we note there is no obvious way to cluster intrinsic goals into meaningful families  (the 13 goal types mentioned in Appendix A.2.1 refer to extrinsic goals from the benchmarks; intrinsic goals are given by $f(s)$ for any state $s$).
> >
> > While we believe a bandit-based mechanism could be made to work, we do not have a clear prediction on whether it would reliably improve upon our simpler fixed-range sampler or justify the additional complexity. We leave this as an interesting direction for future work.

---

> > > ### Comment · Reviewer_QuSP · 2025-11-20
> > >
> > > I think the authors have addressed my questions in a thoughtful and meaningful way. I am happy to read the replies and looking forward for future advances like applying this method in LLM RL training.

---

### Official Review · Reviewer_ompC · 2025-10-31

**Soundness:** 3
**Presentation:** 3
**Contribution:** 3
**Rating:** 6
**Confidence:** 4

**Summary:**

This paper introduces ULEE (Unsupervised Learning of Efficient Exploration), an unsupervised meta-learning method designed to pre-train adaptive reinforcement learning policies. The core challenge it addresses is how to effectively generate and select self-imposed goals for an agent to learn from, particularly for broad distributions of downstream tasks. ULEE's main contribution is an automatic curriculum learning strategy guided by a novel post-adaptation task-difficulty metric. This approach optimizes for an agent's performance after a period of adaptation, rather than its immediate performance. The method combines an in-context learner with an adversarial goal-generation system that finds challenging yet achievable goals, effectively maintaining training at the "frontier of the agent's capabilities". On XLand-MiniGrid benchmarks, ULEE-pre-trained policies demonstrate improved exploration, adaptation, and generalization to novel tasks , resulting in better zero-shot and few-shot performance and providing a strong initialization for longer fine-tuning processes.

**Strengths:**

1. The post-adaptation task-difficulty metric, for me, is novel. It significantly departs from prior works in automatic curricula, which typically evaluate goal difficulty based on the agent's immediate performance. By defining difficulty as the agent's expected success rate after an adaptation budget, the method directly optimizes for the agent's capacity to learn rather than just its current knowledge. This aligns the pre-training objective more closely.

2. The paper originally combines three key components into a single system, ULEE. While concepts like meta-learning, adversarial goal generation, and in-context learners exist, ULEE integrates them synergically. It uses an adversarial "Goal-search Policy" to propose hard goals, a "Difficulty Predictor" network to estimate their post-adaptation difficulty, and an in-context learner (the "Pre-trained Policy") to meta-learn on a curriculum of goals selected for being at the "frontier of the agent's capabilities".

3. The authors evaluate the pre-trained policy across a wide spectrum of downstream scenarios.

**Weaknesses:**

1. The primary weakness of ULEE is its high methodological complexity. The system is not a single algorithm but a complex interplay of four distinct, learning-based components: the Pre-trained Policy ($\pi$), the Goal-search Policy ($\pi_{g, s}$), as well as the Difficulty Predictor. Are there practical bottlenecks (e.g., memory, wall-clock time) for this method?

2. The overall system's success depends on these components learning in lockstep. Can this co-adaptive process be brittle? E.g., will a relatively poor Difficulty Predictor lead to a collapse? Adding a robustness analysis to each design ingredient will greatly strengthen the paper.

3. The pre-training and evaluation tasks (4Rooms-Trivial, 4Rooms-Small, 6Rooms-Small) are all drawn from the same "family" of XLand-MiniGrid rules, which may not fully cover the claim “generalization to new goals, transition dynamics, and grid structures”.


Minors:
1. I recommend a figure to show the overall framework of the proposed, so that it can be more detailed and accessible.

2. There are some grammatical and typographical errors, for example. In line 65, “more tasks become too…”,

**Questions:**

1. There are many hyper-parameters in ULEE, learning rates, network architectures, buffer sizes, sampling bounds LB/UB, number of goal-search episodes etc. How critical are the specific hyper-parameter values?

---

> ### Author Response · Authors · 2025-11-20
>
> Thank you for your detailed and constructive feedback, and for your positive remarks regarding ULEE’s novelty. We address the concerns and questions you raised below.
>
> > W1. *"The primary weakness of ULEE is its high methodological complexity. ... ."*
>
> We agree that ULEE involves several interacting components. However, once the objective is to pre-train with intermediate difficulty goals, we view its design as natural and each component is itself a standard block. Multi-network setups are commonly used in RL (e.g., model-based methods, cooperative and adversarial multi-agent methods, and intrinsic motivation approaches such as RND or empowerment methods), and we view this interplay as part of the contribution rather than a fundamental drawback.
>
> From a practical perspective, the main additional cost comes from the Goal-search Policy, which in our setup adds 25% more environment steps and 15% more PPO updates on top of those of the Pre-trained Policy. Memory usage was dominated by cached Transformer-XL activations, stored transitions, and computation graphs. Since the Pre-trained Policy and Goal-search Policy never run or update simultaneously, there was no practically significant increase in peak GPU memory requirements. The Difficulty Predictor is a small feedforward network that predicts difficulty only for steps taken by the Goal-search Policy; its overall impact on compute requirements is modest and smaller than that of DIAYN’s discriminator or RND’s predictor, which process all collected steps. The goal selection component is a fixed sampling rule (not a learned module) and has negligible compute and memory cost.
>
> Importantly, during deployment only the Pre-trained Policy is used, so memory and wall-clock demands are the same as running any single policy with that backbone. Similarly, during fine-tuning the requirements are essentially the same as running standard $RL^2$.
>
>
> > W2. *"The overall system's success depends on these components learning in lockstep. Can this co-adaptive process be brittle?. ... ."*
>
> > Q1. *"There are many hyper-parameters in ULEE, ..."*
>
> Thank you for highlighting the need for greater clarity regarding robustness and hyperparameter sensitivity. We have addressed these concerns in the general comment and in the new appendix section *“Hyperparameter Selection and Stability”*, which documents the hyperparameters explored and the observed stability across the configurations we tried.
>
> > W3. *"The pre-training and evaluation tasks ... ."*
>
> Goals. During unsupervised pre-training the agent never sees the benchmark extrinsic goals. Environments are sampled from the reward-free distribution $\mu^{\text{unsup}}$, which shares rules and objects with the XLand-MiniGrid benchmarks but does not include their extrinsic goals. Instead, the agent trains only with self-imposed goals generated via $f:\mathcal{S}\to\mathcal{G}$ (either $f_{\text{counts}}$ or $f_{\text{grid}}$); these are not drawn from the 13 extrinsic goal types that define the XLand-MiniGrid task space (Appendix A.2.1). When we say generalization to new goals, we mean that a policy pre-trained solely on self-imposed goals is evaluated on extrinsic goals from the 4Rooms-Trivial, 4Rooms-Small, and 6Rooms-Small benchmarks (Sec 4.3.1 - 4.3.4), as well as on the classical MiniGrid tasks in Sec. 4.3.5, none of which are used during pre-training.
>
> Transition dynamics. The transition dynamics in XLand-MiniGrid environments are determined by sets of rules. The environments used for pre-training ($\mu^{\text{unsup}}$) and evaluation ($\mu^{\text{eval}}$) contain different rulesets. Thus, during evaluation, the agent has to face transition dynamics $p(s'|s,a)$ it has never previously observed. We agree that this generalization is within the same "family" of rulesets. We still believe this axis is worth highlighting since in most RL benchmarks the dynamics are fixed between training and testing.
>
> Grid structures. The generalization to new grid structures claim comes from the experiments in section 4.3.5. During pre-training, the agent only experiences 13x13 grids with four rooms. At evaluation, we test on a suite of MiniGrid tasks with different grid sizes (e.g., 5x5, 8x8, 6x11, 16x16, 19x19) and wall/doors layouts. The Pre-trained Policy’s few-shot performance on these tasks indicates the learned exploration and adaptation strategy generalizes beyond the specific 13x13 four-room structure seen during pre-training.

---

> > ### Author Response · Authors · 2025-11-20
> >
> > **Minors**
> > > *"I recommend a figure to show the overall framework of the proposed, so that it can be more detailed and accessible."*
> >
> > We appreciate the suggestion for adding a figure to show the overall method. We explored several visualizations but ultimately found they either risked omitting key parts of the framework or did not improve clarity beyond the existing explanation in the main text (Sec. 3) and pseudocode (App. A.1). For this reason, we decided to keep the current presentation.
> >
> > > *"There are some grammatical and typographical errors, for example. In line 65, ... ."*
> >
> > Thank you for noting this. We have corrected the error in line 65 and other minor issues. We will continue refining the manuscript as we identify remaining errors.

---

> > > ### Comment · Reviewer_ompC · 2025-11-28
> > >
> > > I read the author's responses, which addressed some of my concerns.  I keep my score.

---

### Official Review · Reviewer_HPvB · 2025-10-31

**Soundness:** 3
**Presentation:** 3
**Contribution:** 3
**Rating:** 4
**Confidence:** 3

**Summary:**

The paper introduces Unsupervised Learning of Efficient Exploration (ULEE) as an unsupervised meta-learning approach that pre-trains a policy capable of rapid adaptation to new tasks. It achieves this by generating challenging-but-solvable goals based on the policy's estimated post-adaptation performance. The resulting policy is an in-context learner that adapts to new goals, dynamics and map structures using only its interaction history (observations, actions, rewards), requiring no explicit goal input. On the XLand-Minigrid benchmark, ULEE outperformed baselines in terms of fast adaptation, and provided a better initialization for both extended fine-tuning and supervised meta-learning. It also demonstrated generalization to novel environment structures.

**Recommendation:**\
This paper falls outside my area of expertise, but appears to have a well motivated and interesting problem setting and strong empirical results. However, I have some questions about the methodology regarding seeding and validation/test sets. Therefore, in its current state, I will recommend to reject. However, I will be open to change my score if my questions are answered satisfactorily.

**Strengths:**

- Although this is not my area of expertise, the paper's motivation and positioning within existing literature appears strong.
- The problem of pre-training for adaptation is very interesting.
- The empirical results are very strong.

**Weaknesses:**

- Section 4.3.1 does not sufficiently answer Q1. In this section the fraction of evaluation goals reached as a function of the number of evaluation episodes is shown in Figure 2. In my opinion this does not isolate exploration as the cause of evaluation goals reached, nor does it answer _"what exploration capabilities"_ the policy exhibits. For example, an increase in evaluation goals reached can also be due to zero-shot generalization, rather than improved exploration/adaptation.
- Some parts of the experimental methodology is unclear. In particular when it comes to hyperparameter tuning and validation vs test split.
- Certain details in the main text could be explained better.

**Questions:**

- Are there better ways to isolate and analyse the exploration capabilities of the Pre-trained Policy? For example, subtract all evaluation goals reached in a single episode and only include the ones reached with more episodes? Or visualize the exploratory behaviour of the policies in some way?
- There appears to be no explicit mention of a test versus validation set. Are the final results evaluated on an independent testing set of environments (that has not been used for validation, hyperparameter tuning, or generally for algorithm design)? Similarly, were the seeds used for the final evaluation (testing) different from the ones used for validation, tuning and design?
- I could not find mention of the hyperparameter tuning approach for your method and the baselines. How is it ensured that your approach did not accidentally benefit from an advantageous hyperparameter combination or tuning budget? Did you use separate seeds for tuning and final evaluation?


**Things to improve that did not impact decision:**
- Some of the related work mentioned in Section 2 is missing an explicit comparison with the paper's approach.
- Line 227: The variable $n$ is introduced there but not defined or mentioned in the text.
- Line 294: I don't quite understand how $f_{counts}$ works.
- Figure 3c: It is unclear to me what exactly Figure 3c is showing. Is it showing post-adaptation return on the evaluation set, evaluated at different points during pre-training?
- Table 1: The bold highlight is very difficult to differentiate from the regular numbers.

---

> ### Author Response · Authors · 2025-11-20
>
> Thank you for your constructive review and for highlighting the paper’s strengths. Below we address your concerns.
>
> **Evaluating exploration**
> > W1. *"Section 4.3.1 does not sufficiently answer Q1. ... ."*
>
> > Q1. *"Are there better ways to isolate and analyse the exploration capabilities of the Pre-trained Policy? ..."*
>
> We agree that the percentage of goals reached with a budget of a single episode can also be used as a measure of zero-shot generalization. They are related, as this zero-shot performance is a direct consequence of the learned exploration strategy.
>
> We note that there is no generally agreed-upon way to measure “exploration capability” in RL. Prior work has relied on various distinct proxies. These include task-agnostic measures such as state coverage, and task-focused metrics such as the number of interactions until specified events of interest occur [1]. Our procedure in Section 4.3.1 follows the latter approach: we treat reaching the diverse goals from XLand-MiniGrid as the set of “interesting events”.  Figure 2 shows how many such events each method can reach under different exploration budgets. This is in line with a common way to evaluate the exploration ability of intrinsic-motivation methods, i.e., by measuring the reward-free performance on sparse goals of interest [2,3].
>
> State-coverage metrics instead judge a method by how many distinct states it visits, independently of whether those states correspond to meaningful tasks. These metrics are particularly popular in domains where the state-visitation distribution can be captured in a single interpretable image (e.g., via a heatmap for an empty grid), which is not our setting.
>
> Moreover, how good an exploration strategy is depends on the task distribution on which it is applied. A good pre-trained agent would then prioritize exploring well on a distribution of human-relevant target tasks, at the cost of worse exploration for other, less relevant tasks. This tradeoff is both unavoidable and essential as environment complexity grows and the space of possible tasks becomes prohibitively large. This is the main reason why we favor task-focused metrics. Since we cannot test over all possible tasks of interest, we use a set of tasks from XLand-MiniGrid as a practical and reasonably broad proxy.
>
> Regarding the suggestion to *“subtract all evaluation goals reached in a single episode and only include the ones reached with more episodes”*, this information is already implicit in Figure 2. For A>B, the difference between the y-values at episode A and B gives the percentage of goals first reached between episodes B+1 and A (your suggestion corresponds to taking B=1). Finally, we remark that during the evaluation procedure the policy never receives any reward. For each environment we simply roll out the Pre-trained Policy and record in how many episodes the sampled XLand-MiniGrid goal is reached. Thus, Figure 2 studies the reward-free behavior learned by the policy.
>
> [1] Colas, C., Karch, T., Sigaud, O., & Oudeyer, P. Y. (2022). Autotelic agents with intrinsically motivated goal-conditioned reinforcement learning: a short survey. Journal of Artificial Intelligence Research, 74, 1159-1199.
> [2] Burda, Y., Edwards, H., Storkey, A., & Klimov, O. Exploration by random network distillation. In International Conference on Learning Representations.
> [3] Hafner, D. Benchmarking the Spectrum of Agent Capabilities. In International Conference on Learning Representations.
>
>
>
> **Experimental methodology**
> > W2. *"Some parts of the experimental methodology is unclear. ... ."*
>
> > Q2. *"There appears to be no explicit mention of a test versus validation set. ... ."*
>
> > Q3. *"I could not find mention of the hyperparameter tuning approach for your method and the baselines. ... ."*
>
> Thank you for pointing out the need for greater clarity regarding fairness and the hyperparameter tuning process. These concerns are now addressed in the general comment and in the newly added appendix section *“Hyperparameter Selection and Stability”*. We hope this clarifies the methodology and resolves your questions.

---

> > ### Author Response · Authors · 2025-11-20
> >
> > **Clarity in explanations**
> >
> > > W3. *"Certain details in the main text could be explained better."*
> >
> > We recognize that the combination of multiple interacting components and evaluation settings in the paper can make the exposition somewhat dense. We aimed to make the descriptions as clear as possible within the available space and appreciate this feedback. If you could point to specific sections or parts that were unclear, we would be grateful, as this would help us further improve the presentation.
> >
> >
> > **Other improvements**
> >
> > > *"Some of the related work mentioned in Section 2 is missing an explicit comparison with the paper's approach."*
> >
> > Due to space constraints and to avoid making the related work section excessively long, we focused on detailing comparisons with what we considered the most closely related approaches, and grouped other methods and introduced them with less depth. We believe this still makes our relationship to prior work clear given the available space.
> >
> > > *"Line 227: The variable $n$ is introduced there but not defined or mentioned in the text."*
> >
> > We have updated the text to introduce it.
> >
> > > *"Line 294: I don't quite understand how $f_{\text{counts}}$ works."*
> >
> > Each grid cell is associated with a corresponding ID that identifies its kind (e.g., wall, red pyramid, empty space). $f_{\text{counts}}$ maps states to a vector that records the number of occurrences of each ID. Thus, an intrinsic goal under $f_{\text{counts}}$ specifies how many instances of each unique object should appear on the grid.
> >
> > We have updated appendix section A.2.1 to clarify this.
> >
> >
> > > *"Figure 3c: It is unclear to me what exactly Figure 3c is showing. Is it showing post-adaptation return on the evaluation set, evaluated at different points during pre-training?"*
> >
> > Yes, this is exactly right. The details are given in the appendix: *“Figure 3.c  tracks performance on $\mu^{\text{eval}}$ throughout pre-training on $\mu^{\text{unsup}}$; at each evaluation point, we sample a fresh set of $1\,024$ environments from $\mu^{\text{eval}}$ and measure the mean performance over the last $5$ episodes of 25-episode interactions.”* We have also updated the figure’s caption to make this more clear.
> >
> > > *"Table 1: The bold highlight is very difficult to differentiate from the regular numbers."*
> >
> > Thank you for pointing this out. We have updated Table 1 to use a stronger bold.
> >
> > ---
> >
> > We thank the reviewer again for raising important questions about our work. Have we sufficiently addressed the reviewer’s main concerns? Please feel free to let us know if there are additional concerns or questions.

---

> > > ### Comment · Reviewer_HPvB · 2025-11-25
> > >
> > > I thank the reviewers for their thoughtful answers. I agree with the arguments laid out by the authors on their approach of measuring the exploration capabilities of the agent (in answer to Q1). My small point of feedback is perhaps more on the specific wording used in Q1. To answer the question _"What exploration capabilities does ... exhibit?"_, I would have expected some clarification on the specific way in which the method explores . Perhaps Q1 could be slightly reworded in terms of _"How efficient are the exploration capabilities ..."_, to align it more with the result-oriented metrics from Figure 2.
> > >
> > > In any case, I consider my questions answered satisfactorily and will raise my score accordingly.

---

> > > > ### Author Response · Authors · 2025-12-01
> > > >
> > > > Thank you for the helpful suggestion regarding the phrasing of Q1 and for your intention to raise the score. We appreciate the feedback.

---

### Official Review · Reviewer_SaiF · 2025-11-01

**Soundness:** 3
**Presentation:** 3
**Contribution:** 3
**Rating:** 6
**Confidence:** 3

**Summary:**

The paper proposes ULEE (Unsupervised Learning of Efficient Exploration), an unsupervised meta-RL pretraining framework that trains an in-context learner via an adversarial goal-generation strategy. Goal difficulty is defined by the post-adaptation success rate, yielding an intermediate-difficulty curriculum.

**Strengths:**

The paper is well-motivated. The curriculum is based on performance after in-context adaptation, not immediate performance, which aligns with the intended meta-RL setting. Empirically, ULEE improves exploration, shows faster few-shot adaptation, and provides stronger initializations for finetuning.

**Weaknesses:**

* The empirical impact of defining difficulty via post-adaptation performance, rather than immediate performance, remains unclear without an ablation. A direct ablation (e.g., a sensitivity study over $K$) would strengthen the paper.
* The baselines do not include recent meta RL and unsupervised RL methods.
* Experimental scope is limited to grid-world domains.

**Questions:**

* Does the learned difficulty correlate with intuitive task hardness? A qualitative or heuristic-based comparison between high and low-difficulty goals would be helpful.
* Does the goal-search policy reliably propose high-difficulty goals? What are the difficulty distributions of goals sampled by the goal-search policy (and a random policy)?
* What other environment information $\xi_M$ could be used, especially for environment domains other than grid-worlds?

---

> ### Author Response · Authors · 2025-11-20
>
> Thank you for your thoughtful review and for recognizing the motivation and empirical contributions of the paper. We address your comments and questions below.
>
> > W1. *"The empirical impact of ... ."*
>
> Thank you for pointing out the need to directly evaluate the impact of using post-adaptation rather than immediate performance to define difficulty. We have addressed this with a new ablation, ULEE (SED), as described in the general comment and in the updated version of the paper.
>
>
> > W2. *"The baselines do not include recent meta RL and unsupervised RL methods."*
>
> Meta-RL baselines are only directly applicable to section 4.3.4 (Fine-tuning with Supervised Meta-Learning). We used the standard $RL^2$ method because it matches how ULEE’s Pre-trained Policy meta-learns, which makes it a particularly fair baseline. While we agree that newer meta-RL methods could have been included, improving the underlying meta-RL algorithm would likely benefit both the baseline and ULEE. A study varying the Meta-RL backbones is an interesting direction for future work but was beyond the scope of this work.
>
> For unsupervised RL, we chose DIAYN due to its popularity as a representative of empowerment-based methods and its direct application to many of our evaluation settings. While our main focus was on the effect of the introduced design choices, we agree that it would be informative to investigate ULEE’s comparison to more recent methods. However, due to the high compute cost of pre-training runs, we leave a broader comparison to future work. For Section 4.3.3 we added RND as a popular intrinsic-motivation-based baseline.
>
>
> > W3. *"Experimental scope is limited to grid-world domains."*
>
> We agree that testing ULEE in additional settings would be valuable. We chose XLand-MiniGrid because it offers large distributions of complex and diverse tasks that capture many of the challenges of more realistic environments while being computationally efficient, and it allows us to test along multiple axes of generalization. Extending and scaling ULEE to richer and broader domains is a natural direction for future work.

---

> > ### Author Response · Authors · 2025-11-20
> >
> > > Q1. *"Does the learned difficulty correlate with intuitive task hardness? ... ."*
> >
> > Our paper defines the difficulty of a goal as 1 minus the success rate over the last K episodes in trying to solve it. A goal is more difficult when the Pre-trained policy fails on it more often, which matches an intuitive notion of what a hard task is. We now give a qualitative comparison when using $f_{\text{counts}}$.
> >
> > Each grid cell is associated with a corresponding ID that identifies its kind (e.g., wall, red pyramid, empty space). $f_{\text{counts}}$ maps states to a vector that records the number of occurrences of each ID. Thus, an intrinsic goal under $f_{\text{counts}}$ specifies how many instances of each unique object should appear on the grid.
> >
> > Consider a rollout of the Goal-search Policy. Initially, as the agent moves without changing the grid, all states $s$ map to the same goal $g_0=f_{\text{counts}}(s_0)$. The Pre-trained Policy solves this goal trivially in one step, since its initial state already satisfies it (it is the same as that of the Goal-search policy). The Difficulty Predictor thus will quickly learn to assign low difficulty to those goals.
> >
> > To find non-trivial goals, the Goal-search Policy must modify the number of times each distinct object is present in the grid. It can do this by either picking up objects or triggering rules (rules define how objects interact). For the Pre-trained policy to succeed in such goals, it must, by trial and error, uncover the same changes the Goal-search policy made. As the number and complexity of changes increases, the more unlikely it is for the Pre-trained Policy to uncover them. This means it will fail more often and the Difficulty Predictor will learn to label those tasks as more difficult. The fact that goals are assigned higher difficulty as the number and complexity of the interactions they require increases, aligns with an intuitive notion of task hardness.
> >
> >
> >
> > > Q2. *"Does the goal-search policy reliably propose high-difficulty goals? ... ."*
> >
> > We tracked the distribution of both the predicted $\hat{d}$ and empirical $\tilde{d}$ difficulty of goals sampled throughout pre-training. It is hard to select representative examples, as these distributions evolve over training and differ across benchmarks and seeds. Instead, we summarize the main qualitative patterns we observed.
> >
> > - The selected goals' difficulty distributions (predicted and empirical) changed most rapidly early in training.
> > - The proportion of goals of intermediate difficulty (both predicted and empirical) was smallest in the later stages of training.
> > - On 4Rooms-Trivial, both distributions became clearly skewed toward easy goals compared to 4Rooms-Small and 6Rooms-Small. In this easier benchmark, the Goal-search Policy struggled to find hard goals by the end of training.
> > - With $f_{\text{counts}}$, the distributions shifted strongly toward hard goals, which came to dominate by the end of training.
> > - Regardless of predicted difficulty, most goals ended up being solved in all or none of the last $K$ episodes (one-sample empirical difficulties of 0 or 1). Goals that were solved only a fraction of the last $K$ episodes were a minority.
> > - Comparing the Goal-search Policy to a random policy, the former resulted in substantially more goals of intermediate and hard difficulty. Under a random policy, the distributions quickly became highly concentrated on easy goals. The random policy only reliably encountered hard goals when using $f_{\text{counts}}$, and even then, almost none were of intermediate difficulty by the last stages of training.
> >
> >
> >
> > > Q3.  *"What other environment information $\xi_M$ could be used, ... ."*
> >
> > The key intuition is that the goal representation alone may not be sufficient to determine the goal’s difficulty. The same goal can be easy or hard depending on the environment (e.g., running at X miles per hour on a track vs on heavy sand). We therefore introduce $\xi_M$ as any information about environment $M$ that helps the Difficulty Predictor estimate how hard a given goal will be in that environment. This could be a state or observation from $M$, trajectories, a textual description of the environment, environment parameters in parametrized settings, or any other accessible information about $M$. None of these are specific to grid worlds, and such descriptors are only used during pre-training, not during deployment or downstream fine-tuning.

---

### Author Response · Authors · 2025-11-20

We thank reviewers SaiF, HPvB, ompC, and QuSP for their time, their constructive feedback, and for highlighting ULEE’s design, motivation, and empirical results. In this general comment, we summarize the main changes in the revised version of the paper.


### **Ablation for use of post-adaptation instead of immediate difficulty**

We thank the reviewers for recognizing the novelty in using post-adaptation instead of immediate performance to guide the curriculum. We also agree with Reviewer SaiF that explicitly ablating this decision would strengthen the paper.
Therefore we have added ULEE (SED) as an ablation (Section 4.2). This variant is identical to ULEE except that it estimates difficulty from the Pre-trained Policy’s immediate performance in single-episode interactions rather than from its performance over the last K episodes of a multi-episode interaction. The new results in Section 4.3.1 and 4.3.2 show that using a post-adaptation metric to guide the curriculum leads to solving more evaluation tasks and yields better zero-shot and few-shot performance, with the gap being larger on the more challenging benchmarks.


### **Hyperparameter selection and stability**

Reviewers HPvB, ompC, and QuSP raised questions regarding hyperparameter tuning, sensitivity, and potential brittleness. In response, we added a new appendix section, *“Hyperparameter Selection and Stability”*, which details our exploration of hyperparameter configurations and the selection process. Across all configurations we tried, we observed no collapse in training and did not find ULEE unusually sensitive to hyperparameter choices. We thank the reviewers for these comments and expect this new section to be useful to future readers.


**Updates to Section 4.3.3 (Fine-tuning on Fixed Tasks)**

We strengthen Section 4.3.3 by (i) adding RND as an additional baseline and (ii) updating the DIAYN results after tuning its fine-tuning hyperparameters. In the original submission, both ULEE (which still does) and DIAYN reused hyperparameters from pre-training. The new DIAYN curve is improved but still below PPO from scratch and ULEE.

---

### Meta-Review · Area_Chair_1hng · 2026-01-09

**Summary:**

The primary concerns regarding this paper is its high methodological complexity and narrow (w.r.t domain diversity or task complexity) experimentation validation. Reviewers acknowledged the novelty of post adaptation difficulty metric but there were concerns about multiple learning based components and choices needed to make it work across environments.

Without extensive benchmarking on standard exploration environments beyond grid worlds, it is difficult to assess whether its performance is the result of a robust algorithmic breakthrough or an artifact of hyper-parameter tuning. However the proposed idea is interesting and should be investigated further. In follow up work, testing in continuous control or high-dimensional sensory environments would help answer the remaining raised questions.

**Reviewer Concerns:**

Addressed:
- Reviewers were concerned about the specific value of defining difficulty via post-adaptation performance. The authors addressed this by adding the ULEE (SED) ablation, which demonstrated that using post-adaptation metrics leads to better few shot performance and coverage.
- Concerns regarding hyper-parameter overfitting were raised by HPvB, ompC, and QuSP. The authors responded by adding a comprehensive hyper-parameter selection appendix section.
- To address the lack of popular non-meta RL baselines, the authors added RND to Section 4.3.3 and updated DIAYN results after further tuning its fine-tuning hyper-parameters.

Outstanding
- The concern that ULEE is only evaluated on grid-world domains (XLand-MiniGrid) remains a point of concern

**Reviewer Scores:**

- SaiF ended with a marginally positive stance after the authors provided the requested SED ablation which proved the empirical advantage of post-adaptation metrics. However, they maintained that the experimental scope remains too narrow due to being limited to grid-world environments.
- HPvB initially recommended rejection but moved to a marginally positive stance after the authors clarified hyper-parameter tuning and evaluation procedures.
- ompC acknowledged the novelty of the post-adaptation metric but remained at a marginal accept due to the high methodological complexity of the four interacting learning components. They noted that the system may still be brittle and that evaluation was limited to a specific family of environment rules.
- QuSP provided a strong accept recommendation and remained satisfied after the authors addressed concerns about adversarial reward hacking and goal selection.

---

### Decision · Program_Chairs · 2026-01-26

Accept (Poster)